nanotechnology/materials science/ nanotechnology

TiO$_2$, quantum dots, microwave–hydrothermal, X-ray diffraction, photocatalysis

**Authors for correspondence:**
Faheem Ahmed
e-mail: fahmed@kfu.edu.sa
Chawki Awada
e-mail: cawada@kfu.edu.sa

# Photocatalytic inactivation of *Escherischia coli* under UV light irradiation using large surface area anatase TiO$_2$ quantum dots

## Faheem Ahmed, Chawki Awada, Sajid Ali Ansari, Abdullah Aljaafari and Adil Alshoaibi

Physics Department, College of Science, King Faisal University, Hofuf, Al-Ahsa 31982, Saudi Arabia

(iD) FA, 0000-0002-5436-1966

In this study, high specific surface areas (SSAs) of anatase titanium dioxide (TiO$_2$) quantum dots (QDs) were successfully synthesized through a novel one-step microwave–hydrothermal method in rapid synthesis time (20 min) without further heat treatment. XRD analysis and HR-TEM images showed that the as-prepared TiO$_2$ QDs of approximately 2 nm size have high crystallinity with anatase phase. Optical properties showed that the energy band gap ($E_g$) of as-prepared TiO$_2$ QDs was 3.60 eV, which is higher than the standard TiO$_2$ band gap, which might be due to the quantum size effect. Raman studies showed shifting and broadening of the peaks of TiO$_2$ QDs due to the reduction of the crystallite size. The obtained Brunauer–Emmett–Teller specific surface area (381 m$^2$ g$^{-1}$) of TiO$_2$ QDs is greater than the surface area (181 m$^2$ g$^{-1}$) of commercial TiO$_2$ nanoparticles. The photocatalytic activities of TiO$_2$ QDs were conducted by the inactivation of *Escherischia coli* under ultraviolet light irradiation and compared with commercially available anatase TiO$_2$ nanoparticles. The photocatalytic inactivation ability of *E. coli* was estimated to be 91% at 60 µg ml$^{-1}$ for TiO$_2$ QDs, which is superior to the commercial TiO$_2$ nanoparticles. Hence, the present study provides new insight into the rapid synthesis of TiO$_2$ QDs without any annealing treatment to increase the absorbance of ultraviolet light for superior photocatalytic inactivation ability of *E. coli*.

## 1. Introduction

Titanium dioxide (TiO$_2$) is a significant nanomaterial which has attracted a considerable attention because of its distinctive

optoelectronic and photocatalytic properties. $TiO_2$ has catalytic, dielectric and optical properties, which leads to diverse industrial applications such as solar cell, pigments, fillers, catalyst supports and photocatalysts [1–5]. Specifically, the $TiO_2$ nanoparticles-based photocatalysis technique is an important and promising method for the complete removal of organic compounds [6,7] and microorganisms [8,9]. In general, the organic compounds can be oxidized to carbon dioxide ($CO_2$), water and simple mineral acids at ambient temperature using $TiO_2$ nanoparticles under the illumination of ultraviolet source [10,11]. Recently, the development of $TiO_2$ and $TiO_2$–Pt catalyst efficiently interacts with the microbial cells under UV light source, showing the microbial cells were completely removed [12]. Moreover, various bacteria, cancerous cells, viruses, algae and fungi were successfully deactivated under the irradiation of UV source using $TiO_2$ nanoparticles [13–17]. Furthermore, Sunada *et al.* [18] reported that the $TiO_2$ nanoparticles are not only killed by the bacteria through photocatalytic process, but also they are used for the decomposition of toxic ingredient of bacteria. If the UV light illuminated for a reasonable time, the bacteria are completely mineralized and converted into $CO_2$, $H_2O$ and other mineral substances [19,20]. On the other hand, the degradation efficiency of $TiO_2$ nanoparticles depends on their morphology, preparation methods and specially size of the particles. The small lateral sized $TiO_2$ nanoparticles exhibit higher specific surface areas (SSAs) [21]. In addition, when the size of $TiO_2$ decreased to below 10 nm, its energy band gap of $TiO_2$ increased due to its quantum size effect [22,23].

$TiO_2$ nanoparticles were prepared by many synthetic routes not limited to but including sol–gel method [24], hydrothermal process [25], template routes [26] and reverse micelles [27]. The sol–gel synthesis process is employed for the controlled synthesis of $TiO_2$ nanoparticles; however, the synthesis of smaller $TiO_2$ nanoparticles with homogeneous size distribution is still challenging. In general, $TiO_2$ nanoparticles prepared by a sol–gel method are amorphous in nature; therefore, a calcination process is required to achieve the crystallinity. Another factor that plays a key role in increasing the photocatalytic activity is the larger SSA of $TiO_2$. Although calcination process could be improved by the crystallinity of $TiO_2$ nanomaterials, it might induce the aggregation of small nanoparticles that leads to a decrease of the SSA. Based on the above concern, we need to synthesize agglomeration-free photocatalytic active $TiO_2$ nanoparticles without any further heat treatment.

Recently, Sofia *et al.* [28] used a novel sol–gel reflux condensation route to produce $TiO_2$ QDs. In their work, the process involved using titanium tetra-isopropoxide as the precursor that was hydrolysed and then subjected to reflux condensation for 24 h. Spherical QD morphology with an average crystallite size of 5–7 nm was obtained by subsequent drying and annealing (450°C for 1 h) treatments. In another report, Xu *et al.* [29] prepared $TiO_2$ quantum dots (QDs) by using an autoclave method, and the mixed solution was heated at 150°C for 24 h in autoclave. They reported that the final product was in the form of mixed structures of monodispersed QDs (3–6 nm) and islands (15–30 nm). Lalitha *et al.* [30] synthesized $TiO_2$ QDs by the sol–gel method. In their work, calcination at 350°C for 30 min was required to obtain $TiO_2$ QDs of 4.8 nm size. Deng *et al.* [31] synthesized anatase $TiO_2$ QDs with surface hydroxyl groups and particle size below 3 nm via a new synthetic route (sol–gel). They have reported that the reaction was completed in a Teflon-lined autoclave, and kept in an oven at 90°C for 1 day.

These reports showed that the preparation methods used for $TiO_2$ QDs are time- and energy-consuming and do not fulfil the economic and industrial requirements of $TiO_2$ QDs-based photo-catalysts. Thus, a simple and fast route, for the synthesis of $TiO_2$ QDs under ambient conditions without any annealing treatment, is still required. Compared with the above-mentioned techniques, microwave–hydrothermal method is much simpler and cheaper due to its unique features such as short reaction time, rapid and homogeneous volumetric heating, enhanced reaction selectivity, energy saving, environment-friendliness and high reaction rate [32].

In this work, we report the synthesis of the agglomeration-free anatase $TiO_2$ quantum dot, for the first time, by using $TiCl_3$ and NaOH by microwave–hydrothermal method toward the photocatalytic deactivation of *Escherischia coli* under UV light source. The microwave-assisted hydrothermal process is adopted to synthesize with controlled size and shape of $TiO_2$ QDs. Most importantly, there is no requirement of further calcination steps to obtain final product as was required in earlier reports [28–31]. The resulting QDs show remarkably high photocatalytic inactivation of *E. coli* as compared with commercially available $TiO_2$ nanoparticles.

## 2. Experimental details

Analytical grade precursors and reagents were used in the present experiments. The synthesis was performed in a microwave–hydrothermal system (CEM-MARS 5). For the synthesis, to prepare aqueous solution, $TiCl_3$

(Sigma Aldrich) and NaOH (99.99%; Sigma Aldrich) in 1 : 10 molar ratio were dissolved in 50 ml deionized water (Milli-Q Gradient A-10 system (Millipore)). The solution was stirred for 20 min at room temperature and transferred into a 100 ml Teflon-lined digestion vessel at 160°C for 20 min with a pressure in the range of 150 psi and 500 W power in a microwave–hydrothermal. After completing the reaction, the solution was cooled down to room temperature. The precipitate was collected and washed several times with water and ethanol. The final samples were dried in an oven at 80°C for 24 h. For the comparison purpose, commercial $TiO_2$ nanoparticles (Anatase, nanopowder, 99.7%; Sigma Aldrich) were used.

X-ray diffraction (XRD) analysis of the samples was carried out using a Phillips X'pert (MPD-3040) X-ray diffractometer with Cu Kα radiation ($\lambda = 1.5406$ Å) operated at a current of 30 mA and a voltage of 40 kV. The morphological studies of QDs were explored through high-resolution transmission electron microscopy (HR-TEM; JEOL/JEM-2100F) operated at 200 kV and a field emission scanning electron microscope (FESEM; MIRA II LMH). UV–Vis spectrophotometer (Agilent-8453) was used to obtain the optical behaviour of the samples ranging from 200 to 800 nm. The optical band gap of the QDs was determined from the UV–Vis diffuse reflectance spectra recorded at room temperature. Raman spectrometer (NRS-3100, $\lambda = 532$ nm) was used to study the structural properties of $TiO_2$ QDs. The SSA of the samples was estimated using Brunauer–Emmett–Teller (BET; Autosorb-1, Quantachrome) analysis.

In photocatalytic experiments, 20, 40 and 60 µg ml$^{-1}$ of aqueous $TiO_2$ solution was prepared through the normal saline water under dark condition. Afterwards, 10 ml of $TiO_2$ solution and 10% fresh standard inoculums of *E. coli* ($\approx 10^8$ cfu ml$^{-1}$) were added into 80 ml sterilized normal saline. For the standardization of the overnight grown culture, where standard inoculum is prepared by diluting and making a 10% inoculum in fresh broth. This fresh 10% inoculum is equivalent to approximately $10^8$ cfu ml$^{-1}$. Before the light exposure, the suspension was stirred with a magnetic stirrer for 30 min in the dark condition. During the dark experiment and irradiation, the beaker was wrapped with an aluminium foil to shield it from the ambient light and to increase reflection. The complete suspension was stirred through the magnetic stirrer, while UV light (Spectronics ENF-240C, ($\lambda = 365$ nm) 4 W tubes) at 15 cm distance from the surface of the medium was illuminated and the suspension was collected every 30 min interval for 4 h. The viable concentration of *E. coli* was estimated with dispersion plate method on nutrient agar. For control, experiment was conducted without the addition of $TiO_2$ into *E. coli* suspension under UV light irradiation. The collected plates were incubated at 37°C for 24 h and the colony counter was used for counting the colonies. For a comparative study, similar concentrations of 20, 40 and 60 µg ml$^{-1}$ of commercial $TiO_2$ nanoparticles solution were used.

# 3. Results and discussion

Figure 1 depicts the XRD patterns of the as-prepared $TiO_2$ QDs and the commercial $TiO_2$ nanoparticles. All the diffractions peaks in $TiO_2$ QDs and commercial nanoparticles are well matched and indexed to anatase phase, and are in good agreement with standard JCPDS card no. 89-4921. From this pattern, the as-synthesized $TiO_2$ QDs exhibit well crystalline peaks with pure anatase, indicating the complete crystallization of the stable anatase phase without any further heat treatment. Mainly, the localized high temperatures through microwaves caused the rapid crystallization of $TiO_2$ QDs [33]. The major diffraction peaks of $TiO_2$ QDs indicate at the same peak position ($2\theta$) as commercial $TiO_2$ nanoparticles. Moreover, the major peak of $TiO_2$ QDs shows broader and the relative peak intensity decreases, which indicates very smaller crystallite size. The average crystallite size (D) of $TiO_2$ QDs and commercial nanoparticles estimated using Debye–Scherrer formula [34] using most intense (101) plane diffraction peaks were found to be approximately 2 nm and approximately 20 nm, respectively.

To identify the morphology and dimension of the $TiO_2$ QDs and commercial $TiO_2$ nanoparticles, FESEM and TEM were used. FESEM images (low magnification) of the $TiO_2$ QDs showed nanoparticles ranging from 2 to 5 nm (figure 2*b*), which can be seen in the high-magnification images as shown in the inset of figure 2*b*. On the other hand, commercial nanoparticles of $TiO_2$ are larger ranging 20–30 nm (figure 2*a*).

Moreover, TEM and HRTEM studies were performed to get the information about morphologies and the structural features of $TiO_2$ QDs. Figure 3 displays the TEM image (low magnification) of the $TiO_2$ QDs of approximately 2 nm size (upper inset of figure 3) which is well matched with the XRD analysis and uniformly distributed (lower inset of figure 3). The HRTEM image (figure 4) displays clear lattice fringes of as-prepared QDs, and it was completely crystalline and entirely consists of an anatase phase. The lattice spacing d is 0.34 nm corresponding to the (101) crystallographic planes of anatase $TiO_2$.

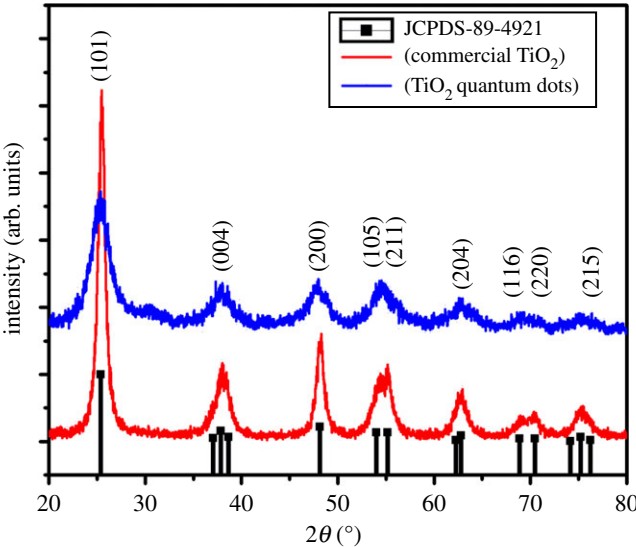

**Figure 1.** XRD patterns of as-synthesized TiO$_2$ QDs, commercial TiO$_2$ nanoparticles and standard JCPDS 89-4921 of TiO$_2$.

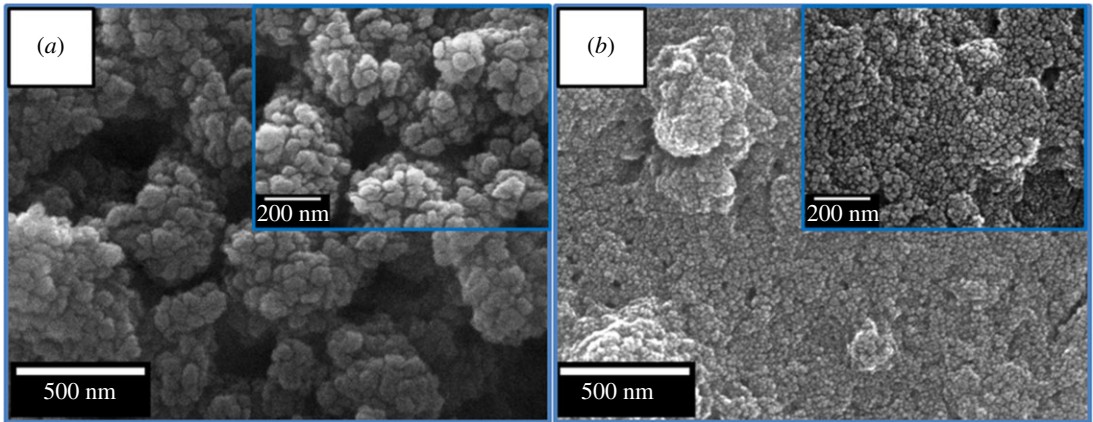

**Figure 2.** Low magnification FESEM images of (*a*) commercial TiO$_2$ nanoparticles, (*b*) TiO$_2$ QDs. Insets of (*a*) and (*b*) show high-magnification FESEM images.

To study the quantum confinement effect of as-prepared TiO$_2$ QDs on the band gap, UV–Vis spectroscopy was employed. Figure 5*a* shows the UV–Vis diffuse reflectance spectra of TiO$_2$ QDs and commercial TiO$_2$ nanoparticles. The band gap energies of the TiO$_2$ QDs and commercial TiO$_2$ nanoparticles were evaluated using Kubelka–Munk function [35,36]. The plot of $(F(R)hv)^2$ versus photon energy ($hv$) for TiO$_2$ QDs and commercial TiO$_2$ nanoparticles is shown in figure 5*b*. The energy band gap of TiO$_2$ QDs was found to be 3.60 eV which is larger than the value of commercial TiO$_2$ nanoparticles as well as the reported value for anatase (3.2 eV) [37]. This increase in $E_g$ might be due to the quantum size effect [38].

Raman spectrum carried out at room temperature further supported the formation of a tetragonal anatase structure of TiO$_2$ QDs confirmed in the XRD. An earlier report [39] showed that for anatase TiO$_2$, six Raman active modes, i.e. A$_{1g}$, two B$_{1g}$ and three Eg, were obtained, and could be detected at 144 cm$^{-1}$ ($E_g$), 197 cm$^{-1}$ ($E_g$), 399 cm$^{-1}$ (B$_{1g}$), 513 cm$^{-1}$ (A$_{1g}$), 519 cm$^{-1}$ (B$_{1g}$) and 639 cm$^{-1}$ ($E_g$). Figure 6 illustrates the Raman spectra of both samples, which indicate the presence of anatase phases TiO$_2$ for both the QDs and commercial nanoparticles. Moreover, the peak corresponding to the B$_{1g}$ mode, A$_{1g}$ and Eg modes of TiO$_2$ QDs shows significant broadening and a small shift toward the higher frequencies than that of the commercial TiO$_2$ (figure 6). It is well known that this shift is attributed to the phonon confinement size effect [40]. In the present work, TiO$_2$ QDs are of approximately 2 nm size; thus, the shift of Raman peaks is due to the quantum size effect.

The SSA of TiO$_2$ plays a key role in photocatalysis [41]. Thus, the primary objective was to prepare larger SSA TiO$_2$ QDs. Figures 7*a* and 8*a* show the nitrogen adsorption–desorption isotherms, and the

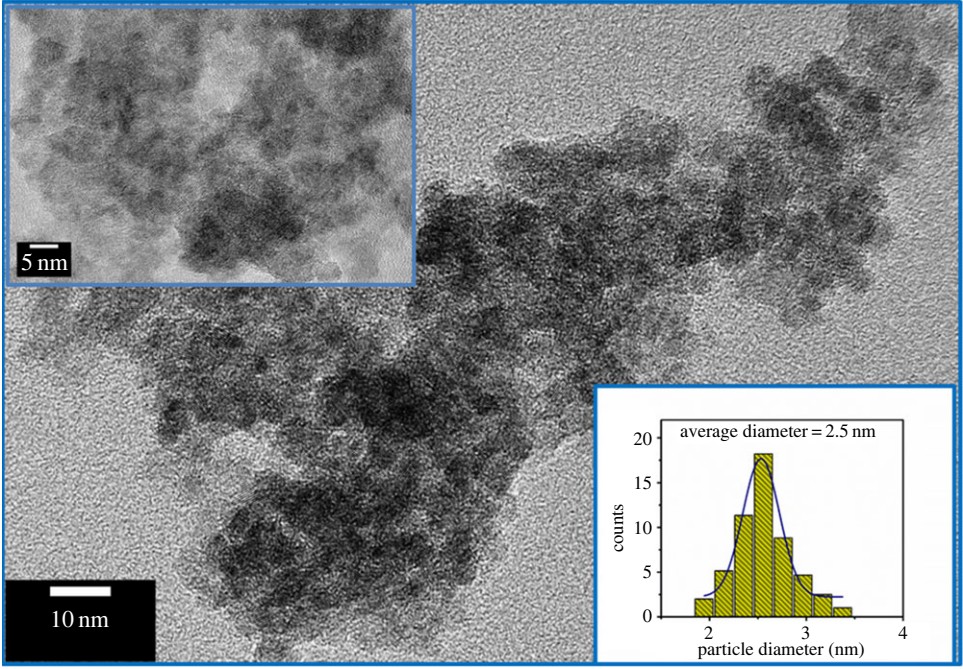

**Figure 3.** TEM image of TiO₂ QDs (low magnification), the upper inset shows high-magnification TEM images and the lower inset shows corresponding particle size distribution plot.

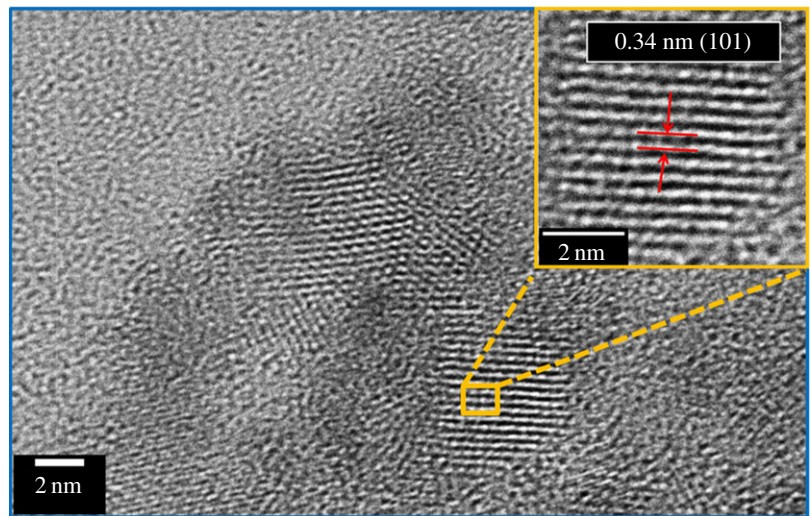

**Figure 4.** HRTEM image of TiO₂ QDs, inset shows high-magnification image of the zoomed area.

SSA plot of the as-synthesized $TiO_2$ QDs and commercial $TiO_2$ nanoparticles are shown in figures 7b and 8b, respectively. The isotherm shows that the nitrogen adsorption volume gradually increases with the relative pressure and then decreases with the decrease of relative pressure (figure 7a). The SSA of the $TiO_2$ QDs was calculated to be approximately 381 $m^2 g^{-1}$ (figure 7b) higher than that of commercial $TiO_2$ particles of approximately 181 $m^2 g^{-1}$ (figure 8b). Also, the $TiO_2$ QDs synthesized by this method showed higher SSA than already reported $TiO_2$ nanoparticles. Yan *et al*. [42] reported preparation of $TiO_2$ nanoparticles with diameter ranging 4–12 nm having an SSA of approximately 64 $m^2 g^{-1}$, on the other hand, Suttiponparnit *et al*. [43] showed an SSA of 254 $m^2 g^{-1}$ of $TiO_2$ nanoparticles. In addition, Lee *et al.* [44] reported $TiO_2$ nanoparticles produced from the sludge of $TiCl_4$ flocculation of wastewater and seawater with average crystallite sizes of 6, 15 and 40 nm with the surface area of 76, 103 and 168 $m^2 g^{-1}$, respectively from artificial wastewater (AW), biologically treated sewage effluent (BTSE) and seawater (SW), respectively. By comparing our results with these reports, the synthesized QDs in this study have a smaller particle size of approximately 2 nm and very large SSA of 381 $m^2 g^{-1}$, thus the presented

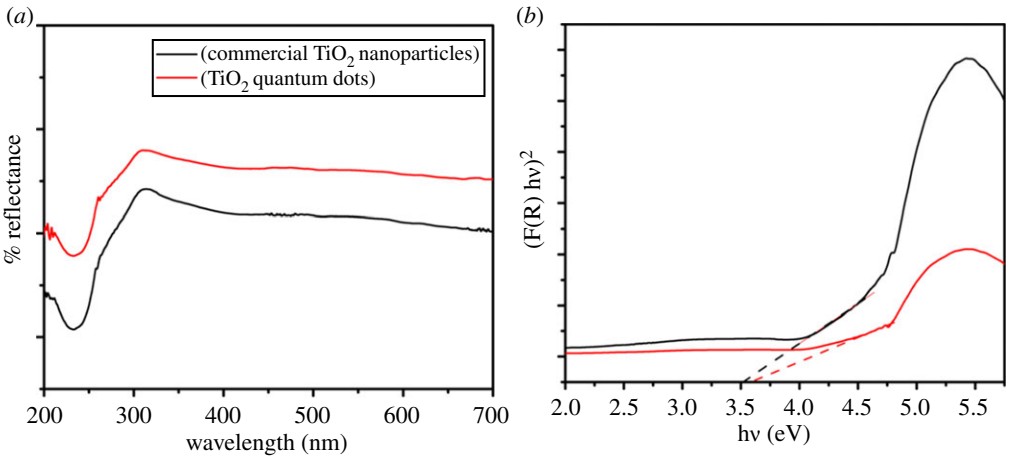

**Figure 5.** (a) UV–Vis diffuse reflectance spectra, and (b) Kubelka–Munk plots for $TiO_2$ QDs and commercial $TiO_2$ nanoparticles.

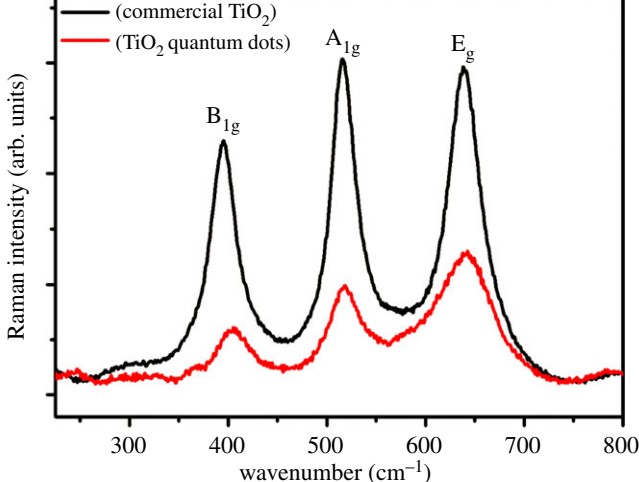

**Figure 6.** Raman spectra of $TiO_2$ QDs and commercial $TiO_2$ nanoparticles.

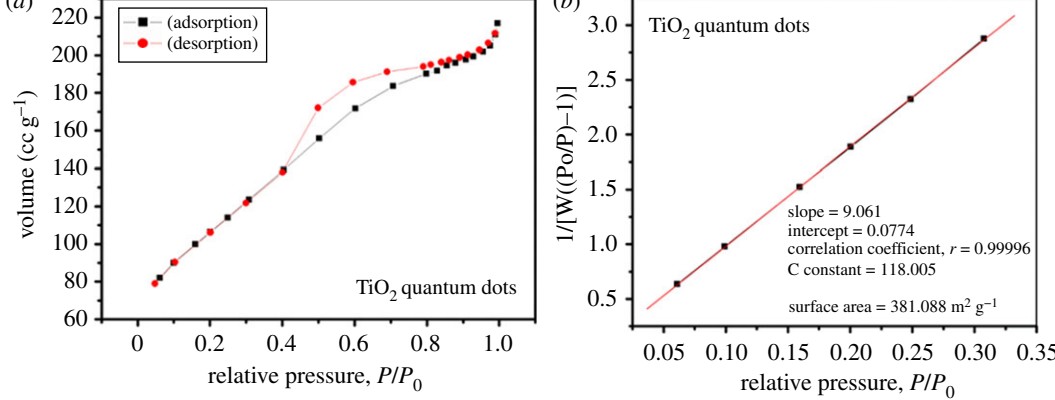

**Figure 7.** (a) Nitrogen adsorption–desorption isotherm of as-synthesized $TiO_2$ QDs, (b) corresponding BET surface area plot.

method is more efficient to produce QDs with large SSA. The higher SSA of $TiO_2$ could enhance the surface reactivity [45].

Furthermore, the photocatalytic deactivation of *E. coli* was conducted by $TiO_2$ QDs and commercial $TiO_2$ nanoparticles powder concentration ranging from 20 to 60 µg ml⁻¹ under UV light irradiation, as shown in figure 9. The nanomaterial will show stronger antibacterial activity if the change occurs in

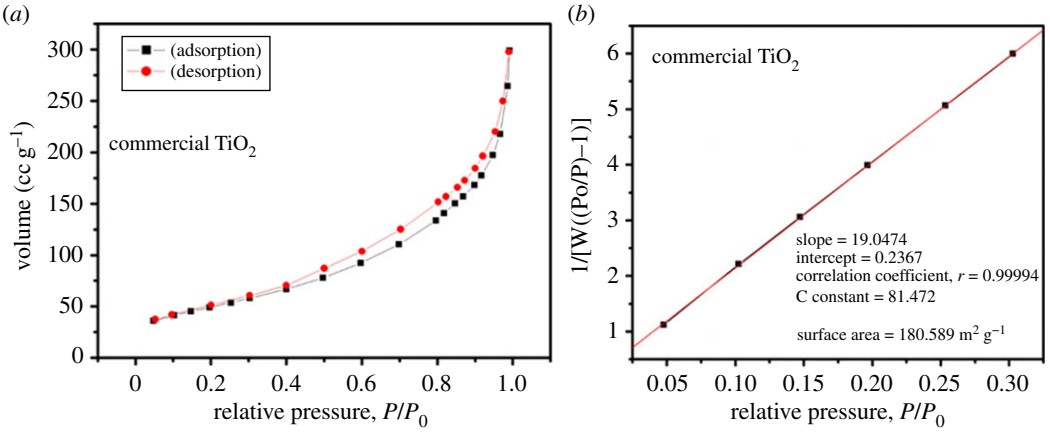

**Figure 8.** (a) Nitrogen adsorption–desorption isotherm of commercial TiO₂ nanoparticles, (b) corresponding BET surface area plot.

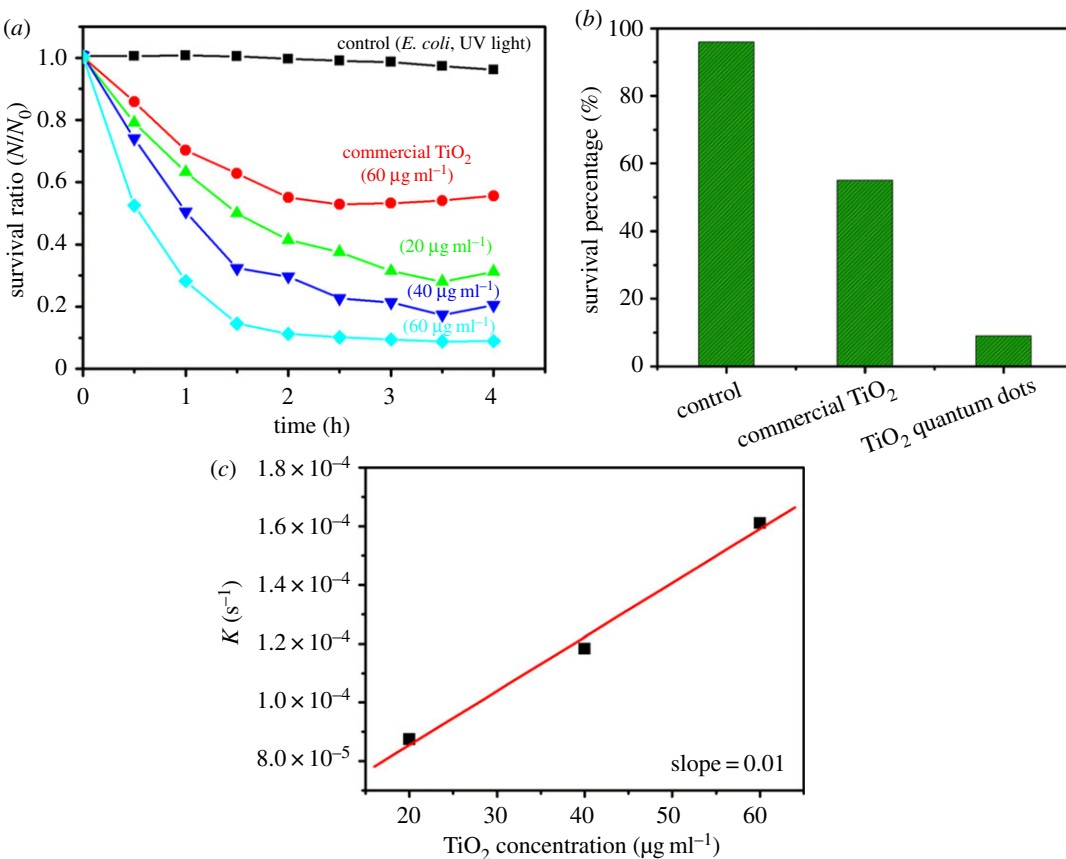

**Figure 9.** (a) Plot of change in the survival ratio of *E. coli*, (b) survival percentage of *E. coli*, (c) the relation between death rate constant and TiO₂ QDs concentration towards *E. coli*.

the survival ratio ($N/N_0$; $N$ = number of cells at time $t$, $N_0$ = number of cell at time $t$ = 0) with the specified time. It is clear from figure 9a that with the increase of UV irradiation time, the survival ratio decreased, which is illustrating the inactivation of *E. coli*. At a specified time, with the increase in powder concentration, the values became smaller, which showed that the higher the powder concentration, the higher the antibacterial activity. In particular, the survival ratio of TiO₂ QDs decreased more steeply in short time as compared with commercial TiO₂ nanoparticles. Figure 9b shows the survival percentage of *E. coli* for control, commercial TiO₂ nanoparticles and TiO₂ QDs. It was observed that the highest inactivation of 91% of *E. coli* was achieved in the presence of 60 µg ml⁻¹ of TiO₂ QDs, while only 45 and 3% of *E. coli* were inactivated in the presence of commercial TiO₂ nanoparticles and control, respectively. It is clear that the TiO₂ QDs show higher photocatalytic inactivation of *E. coli*

than the commercial nanoparticles. By converting the survival ratio of vertical axis into logarithmic value as depicted in figure 9a, a linear decrease for time resulted to the ratio. Thus, death rate constant, k can be determined by first-order kinetics [46];

$$\frac{dN}{dt} = -KN,$$ (3.1)

where N corresponds to survival ratio ($N/N_0$) and t is the time. Figure 9c shows the relationship between K value and TiO$_2$ QDs concentration. The slope value in death rate constant plot of E. coli was found to be approximately 0.01. In comparison with photocatalysis-based antibacterial activity of TiO$_2$ QDs and commercial TiO$_2$ nanoparticles towards E. coli, it was found that the antibacterial activity of TiO$_2$ QDs were much stronger than the commercial TiO$_2$ nanoparticles.

Different activity of the prepared samples is associated with their SSA and the size of the particles. It has been reported that ultra-small particles (i.e. quantum-sized particles) showed better photochemical characteristics than Degussa P25 [47], and have the characteristics between molecular and bulk semiconductor. Thus, there is improvement in the surface-limited reactions due to high surface area-to-volume ratios [45], since TiO$_2$ QDs are purely anatase phase and have extremely large SSA as compared with commercial nanoparticles, which provides better reactivity with the microorganisms and resulted in higher photocatalytic inactivation.

Sunada et al. [48] reported that the photocatalytic mechanism using TiO$_2$ on E. coli is a three-stage process where the decomposition of the dead cell occurred. Fujishima et al. [49] showed that E. coli will be totally mineralized with the illumination time.

In a photocatalytic process, the light with a wavelength greater than or equal to the band gap ($E_g$) of the semiconductor irradiates onto a semiconductor such as TiO$_2$. When the QDs absorb the light, the electrons in the valance band excited to the conduction band, resulting in the generation of photoexcited electron–hole pairs. These photoexcited electron–holes might diffuse to the surface of the semiconductor resulted in the interfacial electron transfer. The oxidation reactions in the solution are caused by holes which resulted in the mineralization of organic substances [50]. In the photocatalytic process, OH$^\bullet$ radicals formed which are governed by OH groups and or physisorbed H$_2$O. The production of highly reactive hydroxyl radicals (OH$^\bullet$) occurred due to the reaction of holes with water, and caused the oxidation of organic materials and biomolecules [51]. To achieve a high efficiency by the adsorption of higher OH groups on the surface of QDs, the large SSA of the TiO$_2$ QDs is a key factor. In another factor the wider band gap of TiO$_2$ QDs prevents the recombination effects of charge carriers, resulting in higher photocatalytic activity against E. coli. Moreover, when the crystallite size of the particle decreases to below or approximately 10 nm, the charge carriers acted quantum mechanically [45]. Due to the confinement, the band gap increased with the decrease of particle size. Thus, with the increase in band gap, the potential of oxidation of the photon-generated holes and the reducing potential of the electrons might increase. Consequently, TiO$_2$ QDs exhibit excellent photocatalytic properties, and this property was used for the inactivation of E. coli.

# 4. Conclusion

In summary, rapid and cost-effective one-pot microwave–hydrothermal route was used to prepare anatase TiO$_2$ QDs within 20 min without any additional heat treatment, and the photocatalytic inactivation of E. coli was investigated. XRD, Raman and HRTEM investigations confirmed the tetragonal anatase structure with well crystalline and single-phase nature. TEM results revealed TiO$_2$ QDs with the size of approximately 2 nm. The BET surface area analysis showed that the anatase TiO$_2$ QDs exhibited a much higher SSA (381 m$^2$ g$^{-1}$) than commercial nanoparticles (181 m$^2$ g$^{-1}$) as well as earlier reported TiO$_2$. The band gap energy for QDs was found to be 3.60 eV higher than that of the commercial nanoparticles. As-synthesized TiO$_2$ QDs exhibit higher photocatalytic inactivation of E. coli than commercially available nanoparticles under UV light irradiation. TiO$_2$ concentration of 60 µg ml$^{-1}$ is sufficient to inactivate about 91% of E. coli. The higher photocatalytic inactivation properties of the TiO$_2$ QDs are believed to be due to smaller particle size and higher band gap resulted in the higher SSA and prevent electron–hole recombination rate compared with the commercial nanoparticles. These QDs effectively inactivate E. coli by photocatalysis and offer an improved charge separation and promote the photoactivity significantly. This work suggests that to obtain excellent photocatalytic properties, tuning of particle size might be a key parameter for promising future biomedical applications.

Data accessibility. Data available from the Dryad Digital Repository: https://dx.doi.org/10.5061/dryad.9pc7mj1 [52].
Authors' contributions. F.A. designed the study and prepared all the samples for analysis. F.A., S.A.A. and C.A. collected and analysed the data. F.A., A.A., and A.A.S. contributed in interpreting the results and writing the manuscript. All authors read the manuscript and gave final approval for publication.
Competing interests. The authors declare no competing interest.
Funding. This work is funded by Deanship of Scientific Research at King Faisal University through NASHER track (grant no. 186101).
Acknowledgements. The authors would like to thank the Deanship of Scientific Research at King Faisal University for supporting this research through NASHER track (grant no. 186101).

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
