## [Reviewer comments · Royal Society Open Science]

Review History

RSOS-190537.R0 (Original submission)

Review form: Reviewer 1

Is the manuscript scientifically sound in its present form?

Yes

Are the interpretations and conclusions justified by the results?

Yes

Is the language acceptable?

No

Is it clear how to access all supporting data?

Not Applicable

Do you have any ethical concerns with this paper?

No

Have you any concerns about statistical analyses in this paper?

No

Recommendation?

Accept with minor revision (please list in comments)

Comments to the Author(s)

The authors describe a simple, hydrothermal synthesis of TiO₂ (anatase) quantum dots without calcining. The synthesis results in very small (2-4 nm) photo-active particles with a demonstrated ability to kill E. coli. The manuscript should be of interest to the community and the data convincingly support the conclusions. I have the following suggestions for revisions, which should be addressed before publishing:

(1) p 2. The authors state that the band-gap “decreases” due to the quantum size effect. This should read “increases,” as is consistent with both the literature their data.

(2) p 2. The authors state that a wider bandgap can prevent recombination of photo-generated electrons and holes in TiO₂. This does not ring true for me: For molecules, at least, transitions of higher energy tend to have shorter lifetimes. Recombination is usually prevented by trap sites in semiconductors. The statement should be supported by a reference or deleted.

(3) Experimental section. The authors should provide more detail about their reagents and grades, including water and NaOH. They should also state the source of the commercial TiO₂ that they used as a comparator and provide the concentration used in the experiment shown in fig. 9(a). I am confused by the description of the E. coli experiments: “Afterwards, 10 mL of TiO₂ solution and 10% fresh standard inoculums of E. coli (~ 10⁸ cfu/mL) were added into 80 mL sterilised normal saline.” Was “10%” meant to read “10 mL”, or does it mean that 10⁸ cfu/mL represents 10% of the 10⁹ cfu/mL standard inoculum, or is it something else? Is 10⁸ cfu/mL the starting concentration or the final concentration? This needs to be clearer in the description. More information is needed about the UV light. What kind of light (e.g. low-pressure Hg), what power (e.g. 300 W), what is the power of the lamp at the target, what is the distance between the lamp and the target? This may not be a complete list; in essence, the authors need to provide enough information so that another person can reproduce the results.

(4) The reflections specified by the JCPDS card should be shown on fig. 1 so that the standard can be compared to the experimental diffractograms.

(5) The authors state that the crystalline peaks in fig. 1 indicate complete crystallisation. Is that so? Could a similar diffractogram not be produced from a mixture of crystallites and amorphous particles?

(6) p 6. The authors state that phonon confinement results in a shift of the Raman peaks to lower frequencies. The shift should be to higher frequencies, and that is what fig. 6 shows, particularly for the 399 cm⁻¹ (E_g) resonance.

(7) There are numerous language errors, but I assume that they will be picked up by the editors before publishing.

Review form: Reviewer 2

Is the manuscript scientifically sound in its present form?

Yes

Are the interpretations and conclusions justified by the results?

No

Is the language acceptable?

No

Is it clear how to access all supporting data?

Yes

Do you have any ethical concerns with this paper?

No

Have you any concerns about statistical analyses in this paper?

No

Recommendation?

Reject

Comments to the Author(s)

The authors present some information about the synthesis of anatase TiO₂ quantum dots by a one-step microwave-hydrothermal method and application in the photocatalytic inactivation of E. coli under UV irradiation. However, there is a lack of research in the formation process of TiO₂ quantum dots, leading to no breakthrough or real novelty in this work, because there are many reports on the application of TiO₂ in sterilization. In addition, some of the statements within the manuscript are not very clear and contradictory between the figure and the results discussion. Therefore, I would not support its publication in Royal Society Open Science. More comments and suggestions are as follows:

- >1. In the abstract, some nouns appear only once and do not require abbreviation, such as SSA, XRD, HRTEM, etc.
- >2. Based on the XRD analysis and HRTEM image, the average crystallite size of TiO₂ quantum dots is ~2 nm. However, the authors mention 'X-ray diffraction (XRD) analysis and high-resolution transmission electron microscopy (HR-TEM) images showed that the as-prepared TiO₂ quantum dots have high crystallinity with anatase phase and size varies from 2 to 4 nm.' in the abstract, it's a contradiction.
- >3. In the introduction, the authors should provide literature reviews about the preparation of TiO₂ quantum dots at present.
- >4. In the introduction, some words are wrong, for example, the 'tome', 'synthetic routs', 'template routs' should be changed to 'time', 'synthetic routes', 'template routes', etc.
- >5. Moreover, the authors mention 'The wider band gap of TiO₂ can prevent the charge recombination effect hence recombination of pair electron hole life time is longer.' in the introduction, please provide strong evidence.
- >6. The authors used microwave-hydrothermal method to prepare TiO₂ quantum dots, we are expected to add the advantages of microwave-hydrothermal method over other methods for preparation of TiO₂ quantum dots.
- >7. For the determination of TiO₂ band gap, the figure should be provided about the relationship between absorption and wavelength.
- >8. In Fig. 6, the authors state that the peak corresponding to the B_{1g} mode, A_{1g} and E_g modes of TiO₂ quantum dots shows a small shift toward the lower frequencies as compared with the

commercial TiO₂, but we find that these peaks are migrating to higher frequencies, the authors should give a reasonable explanation.

>9. To make the results clear, the concentration of commercial TiO₂ nanoparticles for photocatalytic inactivation of E. coli should also be marked in Fig. 9a.

Review form: Reviewer 3

Is the manuscript scientifically sound in its present form?

No

Are the interpretations and conclusions justified by the results?

No

Is the language acceptable?

Yes

Is it clear how to access all supporting data?

No

Do you have any ethical concerns with this paper?

No

Have you any concerns about statistical analyses in this paper?

I do not feel qualified to assess the statistics

Recommendation?

Reject

Comments to the Author(s)

The article "Photocatalytic Inactivation of E. coli under UV Light Irradiation using Large Surface Area Anatase TiO₂ Quantum Dots" was carefully reviewed. The overall outlay and quality of paper is not good for publication in Royal Society of Open Science. Hence, I recommend for its rejection.

Decision letter (RSOS-190537.R0)

30-Apr-2019

Dear Dr Ahmed:

Title: Photocatalytic Inactivation of E. coli under UV Light Irradiation using Large Surface Area Anatase TiO₂ Quantum Dots

Manuscript ID: RSOS-190537

I am writing to you in regards to your manuscript above which you submitted to Royal Society Open Science.

In view of the criticisms of the reviewers and editors, found at the bottom of this letter, your manuscript has been rejected for publication.

Thank you for considering Royal Society Open Science for the publication of your research. I hope this decision will not discourage you from submitting manuscripts in the future.

RSC Associate Editor:

Comments to the Author:

Although Reviewer 1 supported the publication of the manuscript after revisions, because Reviewer 2 and Reviewer 3 (Adjudicator) felt that the manuscript should be rejected and raised a number of technical concerns with the work, your manuscript has been rejected.

RSC Subject Editor:

Comments to the Author:

(There are no comments.)

Reviewers' Comments to Author:

Reviewer: 1

Comments to the Author(s)

The authors describe a simple, hydrothermal synthesis of TiO₂ (anatase) quantum dots without calcining. The synthesis results in very small (2-4 nm) photo-active particles with a demonstrated ability to kill E. coli. The manuscript should be of interest to the community and the data convincingly support the conclusions. I have the following suggestions for revisions, which should be addressed before publishing:

(1) p 2. The authors state that the band-gap "decreases" due to the quantum size effect. This should read "increases," as is consistent with both the literature their data.

(2) p 2. The authors state that a wider bandgap can prevent recombination of photo-generated electrons and holes in TiO₂. This does not ring true for me: For molecules, at least, transitions of higher energy tend to have shorter lifetimes. Recombination is usually prevented by trap sites in semiconductors. The statement should be supported by a reference or deleted.

(3) Experimental section. The authors should provide more detail about their reagents and

grades, including water and NaOH. They should also state the source of the commercial TiO₂ that they used as a comparator and provide the concentration used in the experiment shown in fig. 9(a). I am confused by the description of the E. coli experiments: "Afterwards, 10 mL of TiO₂ solution and 10% fresh standard inoculums of E. coli (~ 10⁸ cfu/mL) were added into 80 mL sterilised normal saline." Was "10%" meant to read "10 mL", or does it mean that 10⁸ cfu/mL represents 10% of the 10⁹ cfu/mL standard inoculum, or is it something else? Is 10⁸ cfu/mL the starting concentration or the final concentration? This needs to be clearer in the description. More information is needed about the UV light. What kind of light (e.g. low-pressure Hg), what power (e.g. 300 W), what is the power of the lamp at the target, what is the distance between the lamp and the target? This may not be a complete list; in essence, the authors need to provide enough information so that another person can reproduce the results.

(4) The reflections specified by the JCPDS card should be shown on fig. 1 so that the standard can be compared to the experimental diffractograms.

(5) The authors state that the crystalline peaks in fig. 1 indicate complete crystallisation. Is that so? Could a similar diffractogram not be produced from a mixture of crystallites and amorphous particles?

(6) p 6. The authors state that phonon confinement results in a shift of the Raman peaks to lower frequencies. The shift should be to higher frequencies, and that is what fig. 6 shows, particularly for the 399 cm⁻¹ (Eg) resonance.

(7) There are numerous language errors, but I assume that they will be picked up by the editors before publishing.

Reviewer: 2

Comments to the Author(s)

The authors present some information about the synthesis of anatase TiO₂ quantum dots by a one-step microwave-hydrothermal method and application in the photocatalytic inactivation of E. coli under UV irradiation. However, there is a lack of research in the formation process of TiO₂ quantum dots, leading to no breakthrough or real novelty in this work, because there are many reports on the application of TiO₂ in sterilization. In addition, some of the statements within the manuscript are not very clear and contradictory between the figure and the results discussion. Therefore, I would not support its publication in Royal Society Open Science. More comments and suggestions are as follows:

- >1. In the abstract, some nouns appear only once and do not require abbreviation, such as SSA, XRD, HRTEM, etc.
- >2. Based on the XRD analysis and HRTEM image, the average crystallite size of TiO₂ quantum dots is ~2 nm. However, the authors mention 'X-ray diffraction (XRD) analysis and high-resolution transmission electron microscopy (HR-TEM) images showed that the as-prepared TiO₂ quantum dots have high crystallinity with anatase phase and size varies from 2 to 4 nm.' in the abstract, it's a contradiction.
- >3. In the introduction, the authors should provide literature reviews about the preparation of TiO₂ quantum dots at present.
- >4. In the introduction, some words are wrong, for example, the 'tome', 'synthetic routs', 'template routs' should be changed to 'time', 'synthetic routes', 'template routes', etc.
- >5. Moreover, the authors mention 'The wider band gap of TiO₂ can prevent the charge recombination effect hence recombination of pair electron hole life time is longer.' in the introduction, please provide strong evidence.

>6. The authors used microwave-hydrothermal method to prepare TiO₂ quantum dots, we are expected to add the advantages of microwave-hydrothermal method over other methods for preparation of TiO₂ quantum dots.

>7. For the determination of TiO₂ band gap, the figure should be provided about the relationship between absorption and wavelength.

>8. In Fig. 6, the authors state that the peak corresponding to the B_{1g} mode, A_{1g} and E_g modes of TiO₂ quantum dots shows a small shift toward the lower frequencies as compared with the commercial TiO₂, but we find that these peaks are migrating to higher frequencies, the authors should give a reasonable explanation.

>9. To make the results clear, the concentration of commercial TiO₂ nanoparticles for photocatalytic inactivation of E. coli should also be marked in Fig. 9a.

Reviewer: 3

Comments to the Author(s)

The article "Photocatalytic Inactivation of E. coli under UV Light Irradiation using Large Surface Area Anatase TiO₂ Quantum Dots" was carefully reviewed. The overall outlay and quality of paper is not good for publication in Royal Society of Open Science. Hence, I recommend for its rejection.

Author's Response to Decision Letter for (RSOS-190537.R0)

See Appendix A.

RSOS-191444.R0

Review form: Reviewer 1

Is the manuscript scientifically sound in its present form?

Yes

Are the interpretations and conclusions justified by the results?

Yes

Is the language acceptable?

Yes

Do you have any ethical concerns with this paper?

No

Have you any concerns about statistical analyses in this paper?

No

Recommendation?

Accept as is

Comments to the Author(s)

The authors have addressed all my concerns.

Review form: Reviewer 2

Is the manuscript scientifically sound in its present form?

Yes

Are the interpretations and conclusions justified by the results?

Yes

Is the language acceptable?

Yes

Do you have any ethical concerns with this paper?

No

Have you any concerns about statistical analyses in this paper?

No

Recommendation?

Accept as is

Comments to the Author(s)

The revised manuscript can be accepted for publication.

Review form: Reviewer 3

Is the manuscript scientifically sound in its present form?

Yes

Are the interpretations and conclusions justified by the results?

Yes

Is the language acceptable?

No

Do you have any ethical concerns with this paper?

Yes

Have you any concerns about statistical analyses in this paper?

Yes

Recommendation?

Accept with minor revision (please list in comments)

Comments to the Author(s)

The entitled article "Photocatalytic Inactivation of E. coli under UV Light Irradiation using Large Surface Area Anatase TiO₂ Quantum Dots" was carefully reviewed. The article is fairly well presented.

It needs revision before considering for publication.

Still lack of literature are missing authors should read below relevant reference and consider for citation in introduction section.

Polyhedron 120, 169-174, 2016

Nanophotocatalysis and Environmental Applications, 139-169, 2020

International Journal of Hydrogen Energy, 2019

<https://doi.org/10.1016/j.ijhydene.2019.07.241>

Green Materials, 1-8, 2019. <https://doi.org/10.1680/jgrma.19.00002>

International Journal of Hydrogen Energy 44 (26), 13022-13039, 2019

Microchemical Journal 147, 7-24, 2019.

English and grammatical errors should be rectified during the revision of the paper.

Decision letter (RSOS-191444.R0)

16-Sep-2019

Dear Dr Ahmed:

Title: Photocatalytic Inactivation of E. coli under UV Light Irradiation using Large Surface Area Anatase TiO₂ Quantum Dots

Manuscript ID: RSOS-191444

It is a pleasure to accept your manuscript in its current form for publication in Royal Society Open Science. The chemistry content of Royal Society Open Science is published in collaboration with the Royal Society of Chemistry.

RSC Associate Editor

Comments to the Author:

Please note that we do not require you to cite the literature suggested by Reviewer 3 as a condition for publication.

Reviewer(s)' Comments to Author:

Reviewer: 1

Comments to the Author(s)

The authors have addressed all my concerns.

Reviewer: 2

Comments to the Author(s)

The revised manuscript can be accepted for publication.

Reviewer: 3

Comments to the Author(s)

The entitle article "Photocatalytic Inactivation of E. coli under UV Light Irradiation using Large Surface Area Anatase TiO₂ Quantum Dots" was carefully reviewed. The article is fairly well presented.

It need revision before considering for publication.

Still lack of literature are missing authors should read below relevant reference and consider for citation in introduction section.

Polyhedron 120, 169-174, 2016

Nanophotocatalysis and Environmental Applications, 139-169, 2020

International Journal of Hydrogen Energy, 2019

<https://doi.org/10.1016/j.ijhydene.2019.07.241>

Green Materials, 1-8, 2019. <https://doi.org/10.1680/jgrma.19.00002>

International Journal of Hydrogen Energy 44 (26), 13022-13039, 2019

Microchemical Journal 147, 7-24, 2019.

English and grammatical errors should be rectified during the revision of the paper.

Appendix A

Reply to reviewer comments on the manuscript- RSOS-190537

We are very grateful to the reviewer for his beneficial and constructive comments, questions as well as suggestions to further improve the quality of the manuscript. We have tried our best efforts to revise the manuscript as per their suggestions and guidelines. The changes done in the manuscript are highlighted by **red color text**. The replies to the reviewer comments are given below:

Reviewer # 1

I have reviewed the authors' answers to my previous comments. I am satisfied with all of them except for comment #2.

Comment # 1

I have read the authors' references 24 and 25 and can find no mention of the influence of band gap on the lifetime of transitions in semiconductors. Pan et al. (2011) state that the slightly larger band gap of the higher-index facets ($\{101\}$ and $\{010\}$) of TiO₂ increases the oxidizing power of holes and reducing power of excited electrons relative to $\{001\}$, but are silent on the issue of bandgap vs time-to-recombination. Batzill (2011) discusses the roles of surface facet, surface modifications, reconstructions, adsorption of monolayers, etc. on photocatalytic activity. I could not find the part of the paper in which the relationship between recombination rate and band gap energy is discussed. Perhaps the authors can direct me to it or provide another reference (and direct me to the passage). Alternatively, they could delete the line.

Answer

We are thankful to the reviewer for this valuable comment. We have deleted the line in the revised manuscript.

Reviewer # 2

The authors have revised the manuscript according to the reviewers' comments. The revised manuscript can be accepted for publication after major revision. But the following contents should be addressed.

Comment # 1

The authors are encouraged to give the high-magnification and low-magnification of TiO₂ quantum dots TEM images.

Answer

We are thankful to the reviewer. High-magnification and low-magnification TEM images of TiO₂ quantum dots have been provided in Fig. 3 in the revised manuscript.

Comment # 2

High-magnification of TiO₂ quantum dots FESEM image should be given.

Answer

High-magnification FESEM image of TiO₂ quantum dots has been provided in the inset of Fig. 2 in the revised manuscript.

Comment # 3

TiO₂ is insoluble in water, the UV-Vis DRS of TiO₂ quantum dots should be given, not the UV-Vis absorption spectra in fig. 5. Eg should be recalculated and analyzed in detail.

Answer

We are thankful to the reviewer for their valuable comments. UV-Vis DRS of TiO₂ quantum dots and commercial nanoparticles have been provided in fig. 5(a) and fig. 5(b). Also, Eg has been recalculated and analyzed in detail in the revised manuscript.

Sincerely yours,

Faheem Ahmed

Physics department, College of Science

King Faisal University, Hofuf, Saudi Arabia